# Studies on Reproductive Development and Breeding Habit of the Commercially Important Bamboo *Bambusa tulda* Roxb

**DOI:** 10.3390/plants10112375

**Published:** 2021-11-04

**Authors:** Sukanya Chakraborty, Prasun Biswas, Smritikana Dutta, Mridushree Basak, Suman Guha, Uday Chatterjee, Malay Das

**Affiliations:** 1Department of Life Sciences, Presidency University, Kolkata 700073, India; sukanya.rs@presiuniv.ac.in (S.C.); prasun.biswas.22@gmail.com (P.B.); duttasmritikana@gmail.com (S.D.); mridushree.rs@presiuniv.ac.in (M.B.); 2Department of Botany, Kalna College, Kalna 713409, India; 3Department of Statistics, Presidency University, Kolkata 700073, India; suman.stat@presiuniv.ac.in; 4Department of Geography, Bhatter College, Kharagpur 721426, India; raj.chatterjee459@gmail.com

**Keywords:** bamboo, sporadic flowering, protandry, pollen germination, cross pollination, self incompatibility, seed setting

## Abstract

Compared to other grasses, flowering in bamboo is quite divergent, yet complex with respect to time to flower, number of individual culms in a population that have been induced at a time (sporadic vs. gregarious), nature of monocarpy, morphology of inflorescences (solitary spikelet vs. pseudospikelet), biology of pollen and nature of genetic compatibility. Wide diversity exists even across species and genotypes. However, due to the rarity of flowering and inaccessibility, few studies have been done to systematically analyse diverse aspects of the reproductive behaviour of bamboo. In this study, four recurrently occurring, sporadic flowering populations of *Bambusa tulda* have been closely observed over the last seven years. Detailed inflorescence and floral morphology and development of reproductive organs have been studied. Pollen viability was assessed by staining and in vitro germination. Self and cross pollination experiments were performed in a plantation site to assess the genetic nature of pollen-pistil interaction. The study identifies interesting reproductive features, that are not common in other grasses. A few important observations include the early appearance of a solitary spikelet vs. late appearance of a pseudospikelet in the flowering cycle, low rate of pollen germination, protandry, self-incompatibility and higher rate of seed setting by the pseudospikelet as compared to the solitary spikelet. The findings will not only be useful to understand the reproductive behaviour of this non-woody timber plant, but will also be useful for forest management and sustainable use of bamboo bioresources.

## 1. Introduction

Bamboos belong to the monocotyledonous plant family Poaceae and subfamily Bambusoideae. They are globally distributed from 51° North to 47° South except in the polar regions [1]. There are ~125 genera and 1670 species of bamboos identified so far [1,2]. Herbaceous bamboos are found in Brazil, Mexico, Paraguay and West Indies; paleotropical woody bamboos are distributed in Africa, India, Japan, Madagascar, Oceania, South China and Sri Lanka; neotropical woody bamboos are found in Argentina, Chile, Southern Mexico and West Indies; whereas north temperate woody bamboos are observed in Africa, India, Madagascar and Sri Lanka [2]. With respect to species diversity, Asia is at the top, followed by South America and Africa [3]. Among Asian countries, China contains ~6.01 million hectares of bamboo vegetation [4], followed by India having with ~160,037 sq km (India State of Forest Report 2019, https://fsi.nic.in/forest-report-2019?pgID=forest-report-2019 accessed between the period of October 2017 to February 2018). Worldwide ethnic populations are heavily dependent on bamboo bioresources, due to their various uses, such as food and beverage, fodder, medicine, fishing gear, handicrafts and artefacts [5,6,7,8].

The flowering behaviour of bamboo is distinctive. Four diverse flowering types have been identified in bamboos. They are (i) sporadic, (ii) massively synchronised (gregarious), (iii) sporadic followed by massively synchronised, and (iv) partial [9,10]. The sporadic flowering events occur in few culms of a population and have been observed in *Bambusa bambos* [11], *Bambusa pallida* [12], *B. tulda* [13], *Dendrocalamus asper* [14], and *Dendrocalamus longispathus* [15]. In contrast, gregarious flowering happens in a synchronised manner in almost every culm growing over a large geographical area and had been observed in many species, such as *Bambusa nutans* [16], *B. tulda* [12,17,18,19,20,21,22], *Dendrocalamus racemosa* [22], *Melocanna baccifera* [21], *Thamnocalamus spathiflorus* [23] and *Thamnocalamus aristatus* [24]. Quite often, sporadic events may be followed by mass flowering in subsequent years and are defined as sporadic-massively synchronised flowering. It has been observed in *B. tulda* [23], *Chusquea culeou*, *Chusquea montana*, *M. baccifera*, *Phyllostachys heteroclada*, *Phyllostachys reticulata* and *Sasa cernua* [10]. Partial flowering events take place in small, discrete populations, and it is neither extended like gregarious, nor restricted like the sporadic type concerning the number of culms flowered. It had been observed in *Pleioblastus simonii* [10]. The flowering time varies between 1–120 years across different species [10]. Another complexity of bamboo flowering is related to the nature of monocarpy, which differs between sporadic and gregarious flowering types. Mass death of the entire population takes place in cases of gregarious flowering, which is not common for sporadic and partial flowering.

Studies of bamboo flowering have traditionally been focused on ecological aspects [2,25,26,27], which have recently moved towards molecular and genetic aspects [28,29,30,31]. In contrast, very few studies have focused on understanding the reproductive behaviour and specialities of bamboo [32,33,34,35]. More studies need to be conducted to understand the reproductive diversity adopted by different bamboo species. In this study, *B. tulda* was selected for many reasons, such as their enormous economic importance, wide distribution, occurrence of diverse flowering types and woody habitats. Four recurrent and sporadically flowering populations of *B. tulda* were observed for seven years to analyse diverse aspects of reproductive development, such as types of inflorescences observed in a flowering cycle, development of reproductive organs, rate of pollen germination, nature of genetic compatibility and amount of seed set.

## 2. Results

### 2.1. Observations on Recurrent, Sporadic Flowering Cycle of B. tulda for Seven Years

The number of flowering clumps (=genet) varied from 1–4 among four studied populations (Table 1; Figure 1). Similarly, the number of flowering culms (=ramet) also varied among the clumps. For instance, 1–8 out of 33–59 culms flowered sporadically for four consecutive years in the case of SHYM7. Whereas, it was 2–8 out of 24–41 culms in SHYM16, 1–17 out of 43–93 culms in BNDL23 and 6–11 out of 29–84 culms in the case of BNDL24 (Table 1).

All these populations were closely observed for seven years to study the flowering cycle. During the initiation of the flowering cycle in spring (February to March, Light 11 h: Dark 13 h), solitary spikelets started emerging in only a few culms of each population (Figure 2). However, by summer, i.e., from April to May (Light 13 h: Dark 11 h), the number of solitary spikelets increased and pseudospikelets started emerging. The maximum number of pseudospikelets emerged from the nodes of flowering branches during July (Figure 2). Subsequently, from August, both solitary and pseudospikelets decreased in numbers and withered by October (Figure 2). Flowering was always followed by the death of the flowered branches, but the flowering culm remained alive until 2-3 recurrent flowering cycles and subsequently underwent senescence.

However, rhizomes of the flowering clump remained active and young culms sprouted from the rhizomes. These sprouted culms attained maximum height before winter (Figure 2). New leaves, as well as branches emerged from old culms from August to October.

### 2.2. Macro- and Micro-Morphology of Solitary Spikelet and Pseudospikelet

Solitary spikelets were observed either on top of the young growing branches or tillers arising from the rhizome of the flowering culm (Figure 3A–D). It was initially bright green in colour and became pale, straw-coloured on withering, 4.5–9.3 cm long, 0.5–1 cm wide. Each solitary spikelet was subtended by a flag leaf, which was smaller in size than other vegetative leaves (Figure 3A,C). They usually developed in a basipetal manner. Mature inflorescences were observed at the top, whereas young ones were located at the base and remained covered by the leaf sheath. The SEM analysis of inflorescence bud revealed a single apical inflorescence meristem (IM, Figure 3D). In contrast, pseudospikelets grew in clusters on the nodes of flowering branches and were devoid of flag leaves (Figure 3E–H). They were 4.3–9.5 cm in diameter and comprised of ~3–34 spikelet units. Here, each inflorescence unit develops on an axis (rachis), which may bear secondary axes (rachilla; Figure 3G). Rachilla bears several bracts. The basal bracts subtended multiple inflorescence buds, while the bracts in the upper region of the rachilla subtended single spikelet units (Figure 3G). The SEM analysis of inflorescence bud revealed multiple inflorescence meristems arranged in a capitate manner (Figure 3H).

Both solitary spikelets and pseudospikelets were composed of indistinguishable spikelet units, which were subtended by 17–50 mm long, distichous, shining, chaffy bracts. The lower most 2–4 florets were reduced to empty glumes, whereas 4–18 fertile florets were located on the top (Figure 4A).

### 2.3. Morphology of Florets and Micro Morphology of Floral Bracts

In a spikelet unit, florets matured in an acropetal manner. Florets were bisexual, 15–19 mm long and 4.5–7 mm broad at the base (Figure 4A–H). Lemmas were broadly acuminate, mucronate, concave, glabrous, many nerved, bright green, 14.5–21.5 mm long and 4.5–6.5 mm broad (Figure 4B). They were overlapping with paleas. Paleas were membranous, penicillate, 2-keeled, 5–7 nerved, 9–14 mm long, 3–5 mm broad at the base and ciliated at the top (Figure 4C). Lodicules were three in numbers, 1.5–2.5 mm long and fleshy. It was they were pale green, waxy, cuneate, oblong, hyaline at the base and whitish fimbriate at the apex (Figure 4D).

Since palea and lemma contain many species diagnostic characters, SEM analyses were performed. The abaxial surface of paleas contained dumbbell-shaped silica cells (SC, Figure 4I) and long cells (LC) with dense prickles in between (Figure 4J). In the middle part of the palea, prickles were higher in number than were SC. On the contrary, SC were dominant in margins (Figure 4K). However, the abaxial surface of the lemma was comparatively smoother. LC were observed, and prickles were lower in abundance (Figure 4K). Both lemma and palea contained trichome-like macro hairs on the margin (Figure 4L). Also, stomatal guard cells were observed in both lemma and palea.

Mature florets of *B. tulda* contained six stamens. Anthers were basifixed, purple or yellow at maturity (Figure 4E). Each anther had two lobes, and the tip of the anther lobes were emarginated (Figure 4E). At the apical part of an anther, a pore-like linear dehiscence suture was observed (Figure 4F). Anther filaments were thread-like and attained 10–17 mm length during anthesis. The ovary was pear-shaped, 2–2.6 mm long; style 2–3 mm long, hairy. Stigma was trifid, 2–3 mm long, having whitish stigmatic papillae (Figure 4G,H).

Studies on developing florets (F1–F9) within a spikelet revealed different rates of developmental progression between androecium and gynoecium (Figure 5A–H). Initially, androecium and gynoecium primordia grew simultaneously in the most immature floret (F1, Figure 5B). Subsequently, differentiation of androecium primordia was initiated in the floret F2 (Figure 5C), while the gynoecium differentiation started in the F3 (Figure 5D). Eventually, the mature anthers protruded ~3 h earlier than gynoecium (F6, Figure 5G). In contrast, gynoecium attained maturity after completion of anthesis and subsequent senescence of anthers (F9, Figure 5H).

### 2.4. Morphology, Cytology and Germination of Pollen Grains

Microscopic observations revealed that pollen grains were globose, mono-ulcerate and their diameter ranged from 22.37–43.2 µm (Figure 6A). SEM analyses revealed that pollen grains were monoporate, with a distinct annulus and granular exine with regulated ridges (Figure 6A). Cytological studies revealed that the dividing microspores exhibited various meiotic stages that included metaphase I, late anaphase I, late telophase I, metaphase II, anaphase II and late telophase II (Appendix A). In many instances, microspores having multiple nuclei were observed. However, no meiotic abnormalities were recorded. To assess the viability of pollen grains, a 2,3,5 triphenyl tetrazolium chloride (TTC) assay was performed. Viable pollen grains appeared red, whereas non-viable ones remained unstained (Figure 6B). Among 2460 pollen grains obtained from *B. tulda*, 602 were viable, and the rate of viability varied from 23.94 ± 4.28% to 25.35 ± 1.87%. This finding was further supported by an in vitro pollen germination assay, although the absolute values were lower in in vitro germination than the TTC assay (Figure 6C–D). This could be due to a higher false positivity rate for the TTC assay. The maximum percentage (14.29 ± 0.8%) of pollen germination was observed in Brewbaker and Kwack’s medium supplemented with 15% sucrose [Appendix A]. Among 396 pollen grains, 57 germinated and the germination percentage varied from 13.48 ± 4.05% to 15.16 ± 5.49% across *B. tulda* populations.

### 2.5. Self-Incompatibility in B. tulda

To understand the nature of genetic interaction between pollen and pistil, the pollen-pistil interaction was studied in vivo in two geographically distant (BNDL 23, BNDL 24; ~340 m) populations of *B. tulda* (Figure 7A–C). The florets started opening at 6:00 a.m., and the maximum number of open florets were observed at 10:00 a.m. AM (Figure 7B). The florets remained open for at least 4 h after anthesis, and the frequency of open florets gradually declined. The rate of pollen germination was almost equal among the four different times investigated. Maximum pollen germination was observed within 2 h post-pollination. Therefore, 10:00 a.m. was selected as the optimum time to perform the self vs. cross pollination experiments. Pollinated stigmas were stained, and attached pollen grains obtained blue colour, whereas the germinated pollen tubes were hyaline (Figure 7D–G). In the case of self pollination, the number of pollen grains germinated was 62 out of 834; whereas, it was 300 out of 802 for cross pollination. Therefore, the percentage of pollen germination was significantly higher (*p* < 0.000) in the case of cross pollination (32.9–41.3%) than self pollination (4.2–8.6%; Figure 7F).

### 2.6. Higher Rate of Seed Setting in Pseudospikelet Compared to Solitary Spikelet Inflorescences

To understand the importance of two different types of inflorescences and their impact on overall seed setting, the total number seeds of obtained from solitary spikelets and pseudospikelets were counted and compared (Figure 8A–C). To avoid any confounding effect, due to the different number of spikelet units present in solitary vs. pseudospikelets on the number of total seeds observed, single spikelet units having 7–9 florets were selected from each inflorescence type. The percentage of seed settings in pseudospikelet was significantly higher at *p* < 0.050 (17.3–25.7%) than solitary spikelets (3.2–9.6%; Figure 8D).

## 3. Discussion

### 3.1. Flowering Time, Cycle and Inflorescence Types: Why So Much Diversity?

In *B. tulda* three kinds of flowering cycles, such as (i) gregarious (massively synchronised), (ii) sporadic, and (iii) sporadic-massively synchronised, had been observed [9,10]. This indicates that even within one species, wide diversity exists with respect to flowering time, which may be regulated by genotype, as well as environment [10,36]. Gregarious flowering generally extends over a large area, while sporadic remain restricted in a few culms. Sporadic events may or may not be followed by mass flowering. When a sporadic cycle is converted into gregarious in subsequent years, it is referred to as sporadic-massively synchronised. This study identified seven sporadic flowering incidents of *B. tulda* over the last seven years (Table 1 and Appendix A). In addition, noticeable differences were also observed with respect to the rate of seed setting and senescence patterns between sporadic and mass flowering [33,37,38]. In mass flowering, the rate of seed production is usually higher, and flowering culms wither after each cycle, which was not the case for sporadic flowering. Since the ecological impact of gregarious flowering is huge, studies have been conducted, and many hypotheses put forward, such as habitat modification, seed predator and resource allocation [26,39]. In contrast, much less attention has been paid to sporadic flowering, and it has mostly been considered an early indicator of future large scale flowering [40]. Our study indicates that sporadic flowering is more frequent in smaller, fragmented populations, and may not necessarily be followed by synchronous flowering.

Apart from the diverse extent of synchrony in flowering time, *B. tulda* also demonstrates diversity in inflorescence morphology and appearance in sporadic flowering (Figure 2). In this study, two types of inflorescences, referred to as solitary spikelet and pseudospikelet, have been identified in *B. tulda* (Figure 3A–H). Although such types of inflorescences were also reported from *Guadua inermis*, *Phyllostachys aurea*, *Dendrocalamus menghanensis* [41,42,43], no studies have aimed to understand their relative abundance in a flowering cycle, particularly with reference to seasonal changes. This study identified that solitary spikelets were more abundant during the early stages of the flowering cycle (March–May) in *B. tulda*, whereas pseudospikelets were more abundant in the later stage (May–July; Figure 2). The differential rate of inflorescence abundance might have occurred, due to the following reasons- firstly, solitary spikelets developed from the apical meristem of a flowering branch, and after their senescence, pseudospikelets emerged in clusters from axillary meristems (Figure 3D,H). Secondly, during the late stage of flowering, the culm initiated to shed off vegetative leaves and mobilised stored photosynthate towards reproductive growth.

### 3.2. Fragmented Flowering Population, Low Pollen Viability and Dispersal Lead to Reduced Seed Setting

The rate of pollen viability was apparently low in *B. tulda*, as evidenced by TTC and the in vitro pollen germination assay. These are effective, yet sensitive methods, which can be affected by many factors, such as temperature, humidity, time of pollen collection, maturity of the florets and composition of assay medium [44,45,46,47]. Although sufficient precautions were undertaken to employ optimum conditions during pollen collection and assay, there could have been issues like time to transport them to the laboratory and small populations frequently subjected to anthropogenic impact, which might also have affected overall pollen quality. Nevertheless, in *Bambusa vulgaris*, pollen germination was only 4.5%, which was linked to chromosomal abnormalities [48]. However, in *B. tulda* no such chromosomal aberrations were observed (Appendix A). In contrast, a very high rate of pollen viability (85%) was observed in *M. baccifera* [48]. Other than viability, successful dispersal of pollen from the anther is also required for efficient dispersal. Also, population size and physical distance between populations are other important factors that determine mating success [35]. All the three populations used in this study were small, consisting of approximately 10–50 clumps and each clump comprising of 7–52 culms. Moreover, anthers of *B. tulda* display narrow apical suture, causing limited liberation of pollen grains (Figure 4F). The optimum mating distance was ~0–50 m for *Dendrocalamus membranaceus* and 0–1500 m for *Dendrocalamus sinicus* [35]. It has been observed that the optimum distance for horizontal dispersal of pollen was ~55 m, and maximum dispersion could be up to ~370 m in maise [49]. The relatively long spatial separation between flowering populations of *B. tulda* (BNDL23, BNDL24; ~340 m) might have resulted in a lower abundance of pollen grains required for successful cross pollination (Figure 7C).

Other factors, such as the activity and abundance of pollinators, also influence the pollination process [33]. In *D. membranaceus* and *D. sinicus*, mating success was not only dependent on clump density, but also on pollinator abundance [33,37,50]. In this study, few insect visitors were observed, suggesting wind pollination as the primary mode of pollen dispersal in *B. tulda*. This is in accordance with previous observations made in Arundinaria gigantea, *Ochlandra travancorica*, *Phyllostachys pubescens*, *D. membranaceus*, *D. sinicus* and *Dendrocalamus strictus* [33,51,52,53,54]. However, insect pollination has been observed in few bamboo species, such as *D. membranaceus*, *D. sinicus*, *Phyllostachys nidularia*, *Guadua paniculata* and *G. inermis* [33,34,55]. The size of pollen grains also influences the rate of pollen dispersal by air. The pollen of *B. tulda* falls into category 3, which includes small pollen having sizes ranging from 33.1–43.9 μm [56], conferring higher buoyancy. Although most grass pollen are known to be air-borne and allergenic, no such observations have been reported for bamboo [57].

A higher rate of seed setting was found to be correlated with flowering area in *Sasa senanensis*, *Sasa kurilensis* and *Sasa palmata*, where the main determining factor was the availability of sufficient pollen [32]. A larger flowering area implies a higher capacity for pollen dispersal. The small size of a population, due to fragmentation is significantly associated with a low seed set [Morgan, 1999]. Availability of pollen grains and pollinators were two primary reasons behind higher amounts of seed setting, due to gregarious flowering in *D. membranaceus* [37] and *Schizostachyum zollingeri* [58]. In contrast, this study and others have recorded a very low amount of seeds in *B. tulda* (Figure 8A,D; [13]). Since the studied populations (BNDL23, BNDL24 and SHYM7) were small and isolated, low availability of pollen might be associated with low seed setting. This reduced seed setting can also be linked to low pollen viability [59]. Similar findings were observed in *B. vulgaris*, where a low rate of pollen viability was responsible for the lack of seed set [44].

Although the total amount of seeds produced, due to sporadic flowering, is low, and their relative abundance is different between solitary spikelets and pseudospikelets (Figure 8B,D). Pseudospikelets were more predominant during the later phase of the flowering cycle, i.e., from May to July, which was right before the promotion of vegetative growth. Therefore, it could be an adaptive strategy for *B. tulda* to ensure reproductive success by maximising seed production preceding the monsoon season.

### 3.3. Protandry and High Rate of Genetic Incompatibility to Ensure Genetic Variability

Since flowering time is unusually extended in bamboo, the occurrence of genetic variability, due to sexual reproduction is less frequent. Therefore, the plant group may adopt diverse morphological and reproductive mechanisms to promote self-incompatibility over self-compatibility. Differential development of reproductive organs, known as protandry and protogyny, increases the possibility of cross pollination. This study identified protandry in *B. tulda*, where the androecium matured before the gynoecium, promoting the primarily outcrossing nature of the plant (Figure 5A–H). This observation was similar to findings in *G. paniculata*, *G. inermis* and *Aulonemia aristulata* [34]. Similarly, protogyny, i.e., earlier development of the gynoecium than the androecium has been observed in other bamboos, such as *D. membranaceus*, *D. sinicus*, *Dendrocalamus sikkimensis*, *P. nidularia*, *P. heteroclada* and *Phyllostachys nuda* [33,55,60]. Even in some rare instances, e.g., in M. baccifera, both protandrous and protogynous flowers have been observed [48].

Another strategy to ensure genetic variability is by maintaining a high rate of cross pollination. In this study, the in vivo pollination experiment performed on two populations of *B. tulda* (BNDL23, BNDL24; ~340 m) suggested predominance of cross pollination over self pollination (Figure 7C–F). Wide variations exist across bamboo species with respect to the predominance of self vs. cross pollination. For example, *M. baccifera* is primarily a cross pollinated species [48]. In contrast, *S. senanensis*, *S. kurilensis*, *S. palmata*, *S. cernua* and *Phyllostachys edulis* and *P. pubescens* are primarily self pollinated [32,61,62]. A few others, e.g., *D. membranaceus* and *D. sinicus*, are predominantly cross pollinated, but may undergo self pollination in unusual situations, such as spatial separation of flowering culms and abundance of pollinators [35]. Consequently, the rate of self pollination is higher in sporadic flowering, whereas it is much less in the case of gregarious one [63,64].

In grasses, the Gametophytic Self Incompatibility (GSI; [65]) is maintained by the two unlinked, multiallelic loci S and Z, whereas in dicotyledonous plants, it is regulated by the single S locus [66]. Since both S and Z loci are multiallelic, their allelic diversity can be an important factor that determines the interaction between pollen and pistil during a compatibility reaction [65,67]. In the case of cross pollinated species, self pollination may occur, due to the unavailability of compatible pollen grains. This may result in the rejection of majority pollen, due to lower allelic diversity during their interaction with the pistil. This may lead to a lower rate of fertilisation, and consequently, low seed set. Although the two loci based gametophytic SI (GSI) is known in grasses, molecular information regarding the genes and their regulation are scanty, and in Bambusoideae they are even rarer. Since diverse mating behaviour is observed in bamboos and an individual species can determine its mating nature based on the availability of pollen grains, pollinators and environmental factors, it would be interesting to know if the S-Z loci based GSI system exists in bamboo.

### 3.4. Semelparous Gregarious Flowering vs. Iteroparous Sporadic Flowering: Ecological Benefits and Costs

Most bamboos are monocarpic, and therefore, culm death is followed by flowering. This has been corroborated by observing the induction of programmed cell death-related genes in *Bambusa arundinacea* [68]. However, the extent of semelparity varies between sporadic vs. gregarious types and even among populations. For example, in the case of gregarious flowering, a single flowering cycle generally persists for two to three years, which is followed by the death of whole flowering populations [14]. This reflects the semelparous nature of gregarious flowering [38]. In contrast, sporadic flowering is predominantly iteroparous, i.e., multiple flowering cycles may recur in a single flowering culm until death [38]. Our observations on sporadic flowering in *B. tulda* revealed that rhizomes of the flowering clump usually remained alive, and new culms may emerge every season (Table 1, Figure 2). In contrast, the death of both culm and rhizome takes place in the case of gregarious flowering, but is compensated by enormous production of seeds. Such mass death causes a sudden decline in forest populations, leading to drastic changes in forest dynamics [69,70,71], due to increased availability of light, deposition of extra organic matters, interactions among species for survival of seedlings (Figure 8C) [69,72,73,74,75]. For example, drastic changes in light intensity after mass death of bamboo culm results in quick growth of new bamboo seedlings along with many tall tree species.

Sporadic flowering may or may not be followed by mass flowering events. Recurrent death of only a limited number of clumps may have much less impact on population dynamics. Yet, it may still cause habitat loss for several endangered species, particularly in fragmented forest areas [26,70]. Additionally, solely sporadic events in *D. strictus* and *D. membranaceus* revealed the consistently low frequency of seed setting [37,60]. However, sporadic events, which resulted in gregarious flowering (sporadic-massive synchronised type), may have a much more severe impact on forest populations [10]. One such study revealed that high rates of seed setting in initial sporadic cycles before the onset of mass flowering potentially initiated regeneration of bamboo population before mass death in *Sasa veitchii* var. *hirsuta* [76]. Such an initial regeneration process may prevent the sudden changes in interaction among the organisms present at diverse trophic levels [77]. It also helps in continuous nutrient cycling and litter production to maintain soil fertility [74].

## 4. Materials and Methods

### 4.1. Population of B. tulda Studied

To study reproductive developments of bamboo, three populations of *B. tulda*, i.e., SHYM7 (Rahuta, Shyamnagar, West Bengal, India, 22.830829° N, 88.405029° E), SHYM16 (Rahuta, Shyamnagar, 22.829591° N, 88.409095° E) and BNDL23 (Rajhat, Bandel, West Bengal, India, 22.934348° N, 88.353255°E, Figure 1), which flowered sporadically were monitored for seven years from 2013–2020 (Figure 2). For the purpose of pollination experiments, BNDL23 and BNDL24 (Rajhat, Bandel, 22.932155° N, 88.355551° E) populations were used. Each population was separated by a distance of at least 300 m and were composed of ~10–50 clumps (=genets) [78], out of which flowering was recorded in 1–5 clump. Each clump was comprised of ~7–52 culms (=ramets), and in few rare incidents >70 culms were present. The height of an individual culm ranged from 15–20 m and diameter from 49–55 mm. All these populations were of mixed type and composed of *B. tulda* along with other bamboo species (~1:3). The number of flowering clumps and culm in each studied population were recorded for seven consecutive years (Table 1).

### 4.2. Studies on Inflorescence and Floral Morphology and Pollen Cytology of B. tulda

To study the morphology of inflorescences and floret, 20 intact inflorescences of each type were obtained from different positions of flowering branches. Inflorescence having ready to open florets were collected from the field in an airtight plastic bag at 6:00 AM in the morning and brought to the laboratory. Numbers of solitary spikelet vs. pseudospikelet and florets per spikelet were obtained from three randomly selected flowering culm per population (Figure 3A–H). Fresh florets and individual floral parts, such as glume, lemma, palea, androecium and gynoecium, were dissected, observed and measured by a stereo zoom microscope (50X, Carl Zeiss, Germany, Figure 4A–I). Florets located at the apex of each spikelet was labelled as F1, and the subsequent florets towards the base were labelled in increasing order. Florets in each of these positions were collected to study the developmental progression of androecium and gynoecium (Figure 5A–H). Pollen grains were collected from anthers during post-anthesis of florets and observed in a bright field microscope (Carl Zeiss, Axiostar Plus, Germany). To study the meiotic cell division in pollen, young spikelets were fixed in Carnoy’s solution between 6:30–7:00 a.m. on the plantation site. The pollen were stained with 2% acetocarmine and were observed under a bright field microscope.

### 4.3. Scanning Electron Microscopy (SEM) Analyses of Inflorescence Buds, Floral Bracts and Pollen Grains

To perform the SEM analyses of inflorescence buds, young buds (>3 mm) of both solitary and pseudospikelets were collected. Outer protective layers were carefully removed to expose the meristem tip prior to SEM analyses (Figure 3D,H). Lemma and palea were obtained from freshly collected florets to observe both dorsal and ventral surfaces (Figure 4I–L). Similarly, pollen grains were collected from anthers during post-anthesis (Figure 6A). Each sample was coated by platinum using POLARON-SC7620, Carbon Accessory (Model-CA76) and were scanned with ZEISS EVO 18 SEM (Carl Zeiss SMT, Germany) having a maximum acceleration voltage of 30 kV.

### 4.4. Test of Pollen Viability by Tetrazolium Staining and In Vitro Pollen Germination Assay

Viability of *B. tulda* pollen was assessed by staining with 2,3,5 triphenyl tetrazolium chloride (TTC) and performing in vitro pollen germination assay (Figure 6B–D). For each of the three populations studied (SHYM7, SHYM16, BNDL24), florets were collected from three randomly selected culms between 8:00–9:00 AM during May, 2015 (~34–37 °C). For each culm, three florets were obtained from randomly selected flowering branches. Anthers from fresh florets were collected during post-anthesis when the anthers were bright yellow or purple. Collectively 18 anthers obtained from 3 florets from each culm were pooled together and were subjected to TTC, as well as in vitro germination assay.

Anthers were immediately kept in micro centrifuge tubes containing ~1.0 mL of TTC solution, incubated in the dark for 30 min and observed under a bright field microscope (Figure 6B,D) [79]. A total of 2460 pollen were used for this analysis. Percentage of pollen viability was calculated by counting stained (viable) vs. non-stained (non-viable) pollen observed in three randomly selected microscopic fields (Figure 6D) and then dividing the number of stained pollen by the total number of pollen and multiplying the proportion, thus, obtained by 100.

For in vitro germination analyses, pollen were collected similarly as described in the case of TTC assay and were immediately dusted in a 0.5 mL pollen germination medium (PGM, Brewbaker and Kwack’s medium). To identify the ideal concentration of sucrose for optimum germination, media were supplemented with 10, 15, 20, 25 and 30% sucrose (*w/v*), incubated for 30 min and observed under a bright field microscope in their natural habitat itself. A total of 1175 pollen were used for this analysis. The optimum percentage of germination was observed in media supplemented with 15% sucrose, and hence, was used for in vitro germination assay. Pollen were incubated in PGM for ~2 h. A total of 396 pollen were observed under a bright field microscope to score numbers of germinated vs. non-germinated pollen (Figure 6C) [80]. Pollen grains were considered as germinated, if the length of pollen tube was greater than the diameter of the pollen grain as per the recommended procedure [81]. Percentage of pollen viability was determined by dividing the number of germinated pollen grains by the total number of pollen grains and multiplying the proportion, thus, obtained by 100 (Figure 6D).

### 4.5. Artificial Pollination Conducted in Situ

To understand the nature of pollen-pistil interaction in *B. tulda*, self and cross pollination experiments were performed in their natural habitat (Figure 7A–C). Two individual flowering populations located in Bandel, West Bengal, India (BNDL23 and BNDL24) separated by ~340 m were selected for this study (Figure 7C). Since bamboos primarily propagate via rhizomes, this experiment was performed in populations, which were distanced by at least 300 m to ensure genetic diversity. However, to ensure maximum pollen viability during artificial pollination, population pairs (BNDL23 and SHYM16), which were distanced by ~33 km. could not be considered (Figure 1). For an individual population, ~7–8 culms were used for this study. Approximately 30 spikelets obtained from each population were used for pollination experiments. This was conducted during March, 2019. Selected florets were emasculated prior to pollination, and pollen were dusted within 15 min to ensure viability. The time for maximum floret opening was monitored for three consecutive days at five different time points (6:00 a.m. AM, 8:00 a.m., 10:00 a.m., 12:00 p.m. PM and 2 p.m.) in both populations (Figure 7B). In addition, to identify the optimum time for pollen germination, pollination experiments were performed at four distinct time points (8:00 a.m., 10:00 a.m., 12:00 p.m. and 2 p.m.; Figure 7A). Pollinated stigmas were collected at 2 and 4 h post-pollination and were immediately fixed in formaldehyde, alcohol, acetic acid solution (FAA, 1:5:0.5). Subsequently, they were treated with 70% alcohol followed by 4 M NaOH and finally washed with distilled water. Pistils were stained with 0.001 mg/mL aniline blue (Merck, Germany) and observed under a bright field microscope (Figure 7D,E). Pollen grains that germinated on stigma were counted. For self and cross pollination experiments, pollination was performed at 10:00 a.m., and the pollinated pistils were collected at 12:00 p.m. In the case of self pollination, pollen from mature anthers obtained from one population were dusted on the stigma of an open flower located on the same clump (geitonogamy). On the contrary, cross pollination was performed by dusting pollen collected from one population to the stigma in another population (xenogamy). Immediately after pollen dusting, pollinated flowers were covered with plastic bags to prevent any subsequent pollination. The percentage of germination was determined by dividing the total number of germinated pollen to the total number of pollen and multiplying the proportion, thus, obtained by 100 (Figure 7F).

### 4.6. Seed Numbers Obtained from Solitary Spikelet vs. Pseudospikelet

To compare between the number of seeds obtained from solitary spikelet and pseudospikelet, both the inflorescences were collected from ~7–12 culms from each of the three populations from March to July, 2018 (SHYM7, SHYM16, BNDL23, Figure 8A–D). Seeds were counted from 96 solitary spikelets and 99 spikelet units obtained from 120 pseudospikelet. Both solitary and pseudospikelets were randomly selected from the flowering branches of ~6–7 culms from a clump. To make the analysis comparable, only spikelets containing 7–9 florets were selected from both the inflorescence types. In the case of solitary spikelet, the total number of seeds produced by all the florets in that inflorescence was counted. On the other hand, for pseudospikelet, only the fully developed mature spikelets with a similar number of florets located in that cluster were selected, and the total number of seeds were counted.

### 4.7. Statistical Analyses

For pollen viability assessed by staining, as well as in vitro pollen germination assay, a Pearson’s chi-squared test was performed to analyse whether the differences in proportion were statistically significant across *B. tulda* populations. A two-sample t-test of mean was performed to test whether seed setting percentage for solitary spikelet and pseudospikelet were significantly different. Similarly, a two-sample approximate Z-test t-test was performed to analyse the statistical difference between the proportion of pollen germination in the case of self vs. cross pollination for *B. tulda*.

## Figures and Tables

**Figure 1 plants-10-02375-f001:**
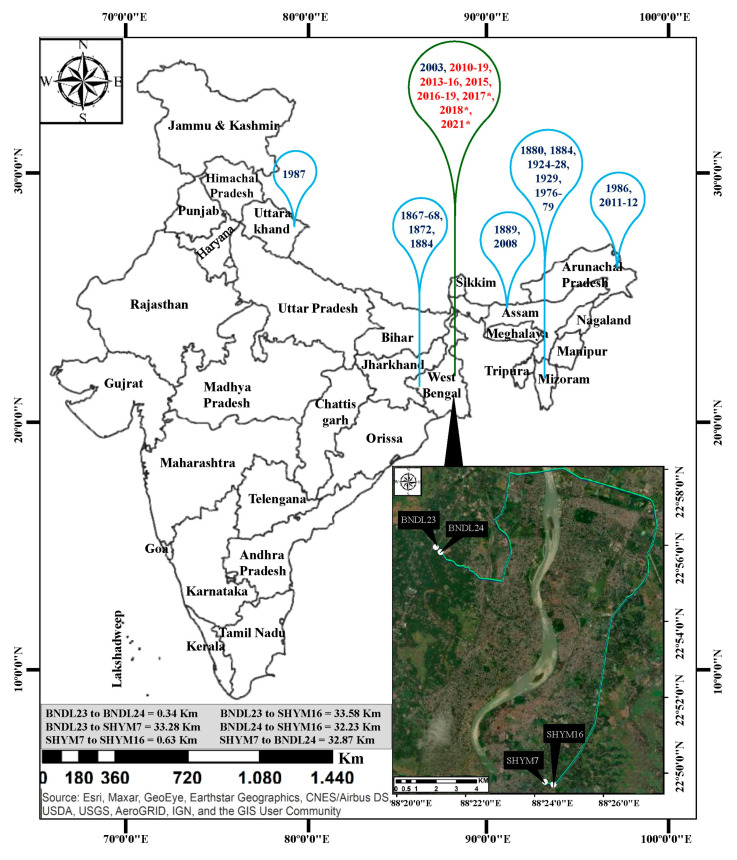
Gregarious and sporadic flowering incidents of *B. tulda* previously reported from different regions of India and the study sites used in these analyses. Data sources for Map: Esri, Maxer, GeoEye, Earthstar Geographics, CNES/Airbus DS, USDA, USGS, AeroGRID, IGN and the GIS user Community. Blue bubbles represent gregarious flowering, and green bubbles represents sporadic flowering events (Troup, 1921; Perry, 1931; Mohan Ram and Gopal, 1981; Rawat, 1987; Gupta, 1987; Naithani, 1993; Bhattacharya et al., 2006; Sarma et al., 2010; Naithani et al., 2013). Those with a flowering cycle that continued until June, 2021, have been marked with asterisks. Flowering year marked in red font identified in this study.

**Figure 2 plants-10-02375-f002:**
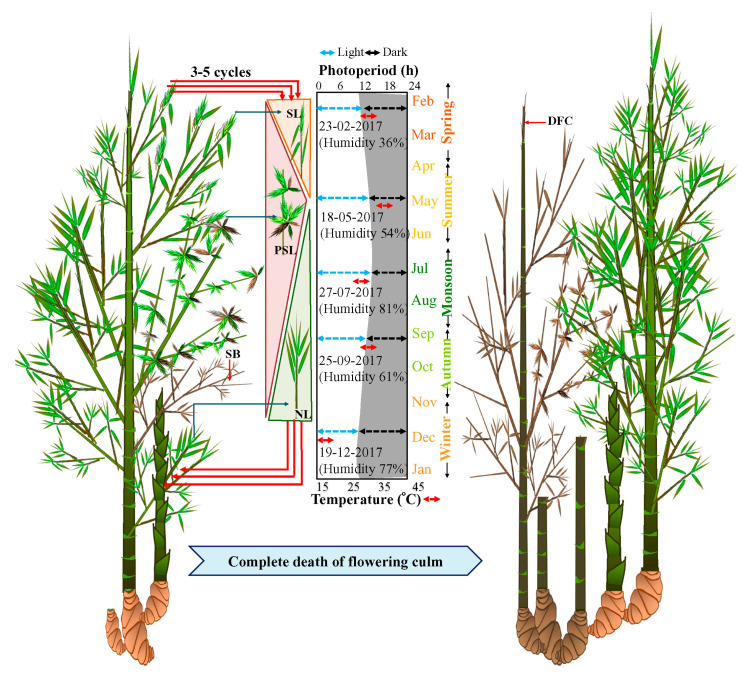
Induction of sporadic flowering and seasonal effect on emergence of solitary spikelet and pseudospikelet in *B. tulda*. Abbreviations used: SB—Senesced flowering branch, SL—Solitary spikelets, PSL—Pseudospikelets, NL—New leaf, DFC—Dead flowering culm.

**Figure 3 plants-10-02375-f003:**
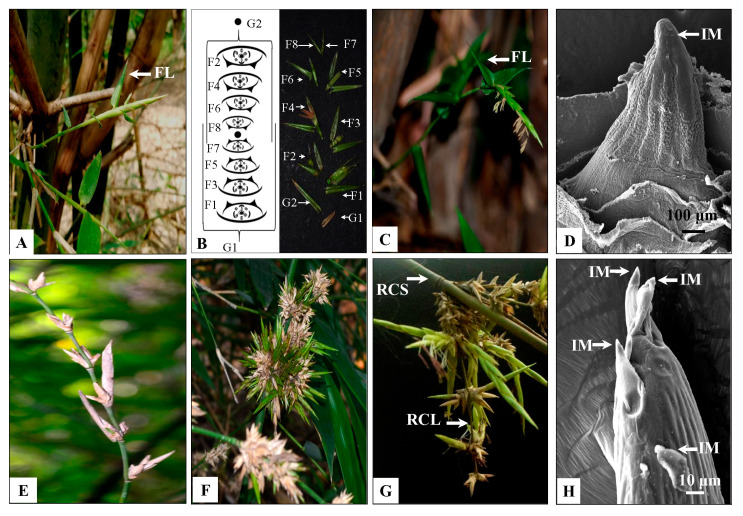
Morphological observation and Scanning electron microscopy (SEM) of the inflorescence of *B. tulda*. (**A**) Solitary spikelet from the lateral branch. (**B**) Organisation of a solitary spikelet. (**C**) Solitary spikelet from a rhizome as a tiller. (**D**) SEM images of spikelet with single inflorescence meristem. (**E**) Pseudospikelet. (**F**) Pseudospikelet in a capitate form. (**G**) Rachis (RCS) and rachilla (RCL) of pseudospikelet. (**H**) SEM images of pseudospikelet with multiple inflorescence meristem. Abbreviations used: FL—Flag leaf, IM—Inflorescence meristem.

**Figure 4 plants-10-02375-f004:**
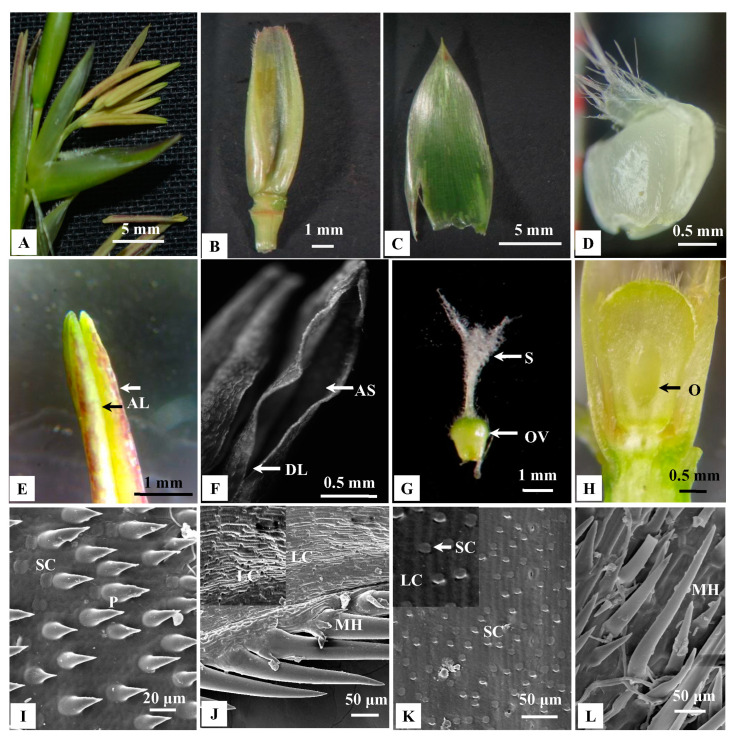
Morphological observation of different floral parts of *B. tulda* and Scanning electron microscope (SEM) images of the abaxial epidermis of palea and lemma; (**A**) Single floret with six basifixed anthers. (**B**) Dorsal view of a lemma. (**C**) Dorsal view of a palea. (**D**) Lodicule. (**E**) Anther showing two lobes. (**F**) Apical suture of the anther. (**G**) Gynoecium with trifid stigma. (**H**) Longitudinal section of ovary. (**I**) Presence of prickles (P) and silica cells (SC). (**J**) Presence of macro hair (MH) and long cells (LC; magnified view inset). (**K**) Presence of SC and LC, (LC; magnified view inset). (**L**) Presence of macro hairs (MH). Abbreviations used: F—Floret, G—Glume, AL—Anther lobes, AS—Apical suture, DL—Dehiscence line, OV—Ovary, O—Ovule, MH—Macro hair, LC—Long cells, SC—Silica cells, P—Prickles.

**Figure 5 plants-10-02375-f005:**
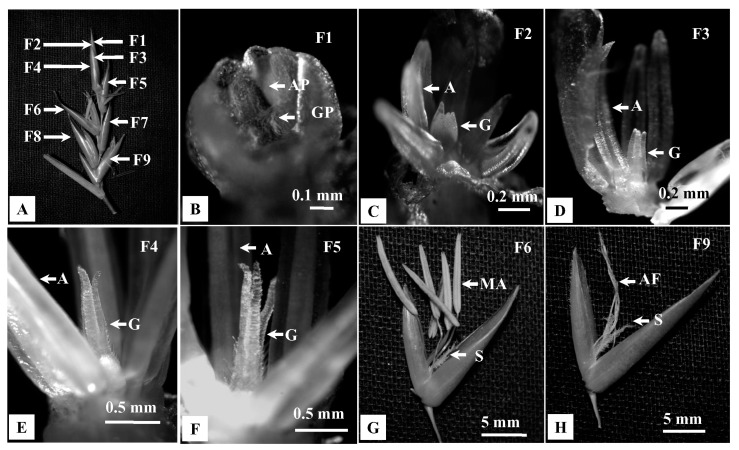
Developmental progression of androecium and gynoecium in florets of *B. tulda*. Floret numbers are marked from apex to base in order of maturity (F1—youngest, F7—oldest floret). (**A**) Arrangement of florets in a spikelet. (**B**–**E**) Developmental stages of androecium and gynoecium primordia. (**F**) Unopened flower with matured anthers and immature gynoecium. (**G**) Open floret with matured anthers and immature stigma. (**H**) Floret with anther filaments and developed stigma. Abbreviations used: AP—Androecium primordia, GP—Gynoecium primordia, A—Androecium, G—Gynoecium, MA—Mature anther, AF—Anther filament, S—Stigma.

**Figure 6 plants-10-02375-f006:**
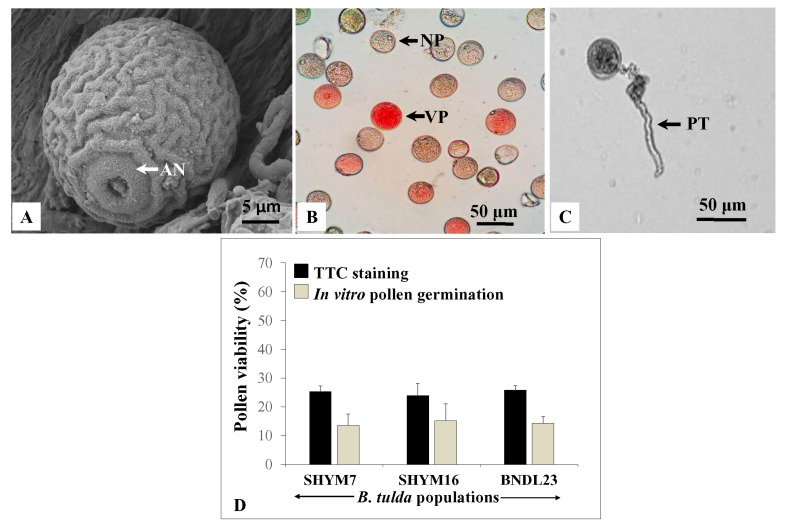
Studies on pollen morphology and viability. (**A**) SEM image of a monoporate pollen having granular exine. (**B**) Pollen grains stained using 2,3,5 triphenyl tetrazolium chloride (TTC) and (**C**) Germinated pollen grain. (**D**) Bargraph of the pollen viability based on TTC staining (black bar) and germination assay (grey bar). Pearson’s chi-squared test revealed that the differences in proportion were statistically not significant across *B. tulda* populations for both TTC and germination assay. Abbreviations used: AN—Annulus, NP—Non-viable pollen, VP—Viable pollen, PT—Pollen tube.

**Figure 7 plants-10-02375-f007:**
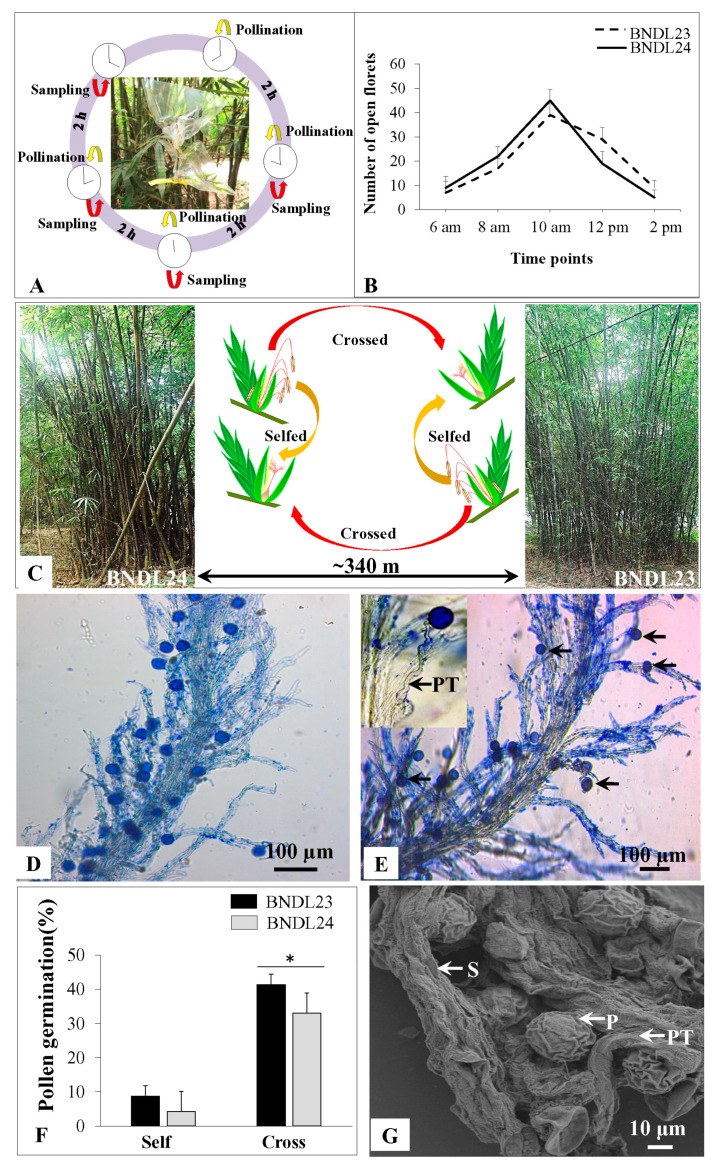
Experimental design and microscopic observation to study *in vivo* pollen-pistil interaction in *B. tulda*. (**A**) Time course experiment was conducted to identify the optimum time point for pollination. (**B**) Line graph demonstrating the optimum time point for opening of the *B. tulda* flower. (**C**) Schematic representation of the experimental set up. (**D**) Magnified view of stained stigma after self pollination. Pollen tubes are marked by an arrow. (**E**) Magnified view of stained stigma after cross pollination. Pollen tubes are marked by an arrow. (**F**) Histogram demonstrating the relative abundance of self vs. crossed pollination in two populations of *B. tulda*. Two-sample approximate z-test was performed to test statistical significance at *p* < 0.000. While performing the test of significance, data for cross pollination obtained from BNDL23 and BNDL24 were pooled and compared to the pooled data obtained for self pollination. (**G**) Scanning electron microscopic (SEM) image of stigma after cross pollination. Abbreviations used: h—Hour, S—Stigma, PT—Pollen tube, P—Pollen.

**Figure 8 plants-10-02375-f008:**
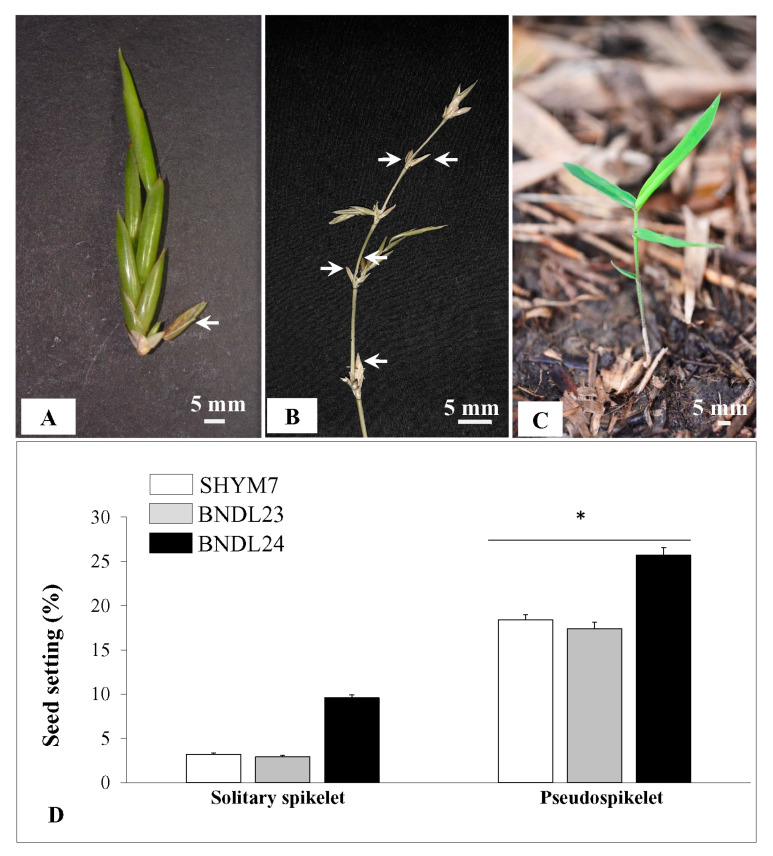
Studies on the rate of seed setting between solitary spikelets and pseudospikelets. (**A**) Solitary spikelet inflorescence with single seed-bearing floret. (**B**) Pseudospikelet inflorescence with multiple seed-bearing florets. Seeds are marked by an arrow. (**C**) Newly emerged young seedling. (**D**) Histogram demonstrating differential seed set observed between solitary spikelets vs. pseudospikelets. Two-sample two-sided *t*-test was performed to test statistical significance at *p* < 0.050. While performing the test of significance, data for pseudospikelets obtained from the three populations were pooled and compared to the pooled data obtained for solitary spikelets.

**Table 1 plants-10-02375-t001:** Comparison between numbers of flowering vs. non-flowering clump and culm observed for seven years in four populations of *B. tulda* studied. Absence of flowering was indicated by (--).

Population Number	Duration of Flowering	Clump and Culm	Year 1	Year 2	Year 3	Year 4	Year 5	Year 6	Year 7
SHYM7	2013–16	Number of flowering clumps	1	1	2	1	0	--	--
		Total culm number in a flowering clump	42	45	59	33	--	--	--
		Number of flowering culms	3	5	8	1	--	--	--
		Number of withered culms	0	1	2	7	--	--	--
SHYM16	2016–19	Number of flowering clumps	2	2	2	1	0	--	--
		Total culm number in a flowering clump	38	41	40	24	--	--	--
		Number of flowering culms	7	8	5	2	--	--	--
		Number of withered culms	0	0	5	9	--	--	--
BNDL23	2013–19	Number of flowering clumps	4	4	3	3	3	3	1
		Total culm number in a flowering clump	77	93	86	87	79	55	43
		Number of flowering culms	11	14	17	15	6	5	1
		Number of withered culms	2	5	9	6	11	12	8
BNDL24	2017–to date	Number of flowering clumps	1	1	2	1	2	--	--
		Total culm number in a flowering clump	31	33	71	29	84	--	--
		Number of flowering culms	6	10	9	8	11	--	--
		Number of withered culms	1	0	5	2	1	--	--

## Data Availability

The data presented in this study will be available on request from the corresponding author.

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
