# Peer review of "Studies on Reproductive Development and Breeding Habit of the Commercially Important Bamboo Bambusa tulda Roxb"

_plants, 2021, doi:10.3390/plants10112375_

Round 1

Reviewer 1 Report

General comments:

This study represents a complete reproductive biology analysis of Bambusa tulsa. Although this work is a bit descriptive, it covers an impressive range of techniques to study phenology, pollen biology, and morphological description of the reproductive cycle of this species. Then, I encourage the publication of this work in this journal after several minor comments and suggestions which the authors could address or consider. I also positively consider the large amount of graphical material in figures that permits the results to be understood well.

Specific comments:

  1. Introduction (line 45): What is the meaning of "the flowering behavior of bamboo is distinc, yet divergent? I do not understand well this statement.
  2. Introduction (line 46): Search better another term for 'flowering incidents'.
  3. Introduction (lines 46-48): Expose better the four types of flowering behaviours. What is the difference between sporadic and gregarious flowering events, the number of specimens flowering within a population?
  4. Results (line 112-117): Explain the morphological and reproductive differences between spikelets and pseudospikelets in Bamboo species.
  5. Discussion (line 261): I think "lowering" should be "flowering".
  6. Discussion (lines 253-263): Sporadic vs. gregarious flowering patterns are compared, but what is the reproductive advantage of one against the other? Is it known or previously published?
  7. Discussion (line 268): Phyllostachys aurea in italics.
  8. Discussion (lines 284-291): Here there is an important issue which should be discussed profoundly. I am not experienced in bamboo species, but they appear insect-pollinated plants according to the next paragraphs. Taking into account the size of the pollen grain, it could be also dispersed by the air (Romero-Morte et al. 2018). Romero-Morte et al. compared different pollen sizes of grasses and the capability to be abundant in the air as aeroallergen. Clinical aspects do not appear to be one of the aims of this study, but it is important to discuss more about the pollination strategies of this species.

Romero-Morte, J., Rojo, J., Rivero, R., Fernández-González, F., Pérez-Badia, R., 2018. Standardised index for measuring atmospheric grass-pollen emission. Science of The Total Environment 612, 180–191. https://doi.org/10.1016/j.scitotenv.2017.08.139

  1. Discussion (lines 317-340): I really liked the way to demonstrate protandry and the reproductive advantage of cross-pollination in this species. Also, the results and the way of showing it in the manuscript.
  2. Materials and methods (lines 386-426): Please, provide more information about how pollen grains are treated and collected, and if the authors used several replicates for each analysis to maintain the representativeness (Rojo et al. 2015). It should be clearly remarked in this section as it is relevant from a statistically point of view.

Rojo, J., Salido, P., Pérez-Badia, R., 2015. Flower and pollen production in the Cornicabra olive (Olea europaea L.) cultivar and the influence of environmental factors. Trees 29, 1235–1245. https://doi.org/10.1007/s00468-015-1203-6

Author Response

Reviewer 1

1. Introduction (line 45): What is the meaning of "the flowering behavior of bamboo is distinct, yet divergent? I do not understand well this statement.

Response: We thank Reviewer 1 for pointing this out. As per the suggestion, the sentence has been revised now in the updated version of the manuscript. (page- 2, line- 49, yellow highlighted).

2. Introduction (line 46): Search better another term for 'flowering incidents'.

Response: We agree with the reviewer and , the term 'flowering incidents' have now rephrased into ‘flowering type’, which is more appropriate  (page- 2, line- 50, yellow highlighted).

3. Introduction (lines 46-48): Expose better the four types of flowering behaviours. What is the difference between sporadic and gregarious flowering events, the number of specimens flowering within a population?

Response: We thank Reviewer 1 for this valuable comment. Yes, the sporadic flowering events occur in few culms of a population while, gregarious flowering happens in a synchronized manner in almost every culm growing over a large geographical area. We now have explained it more explicitly. In addition, we have defined all the four types of flowering observed in bamboo in the introduction section of the revised manuscript (page- 2, lines-50-65).

4. Results (line 112-117): Explain the morphological and reproductive differences between spikelets and pseudospikelets in Bamboo species.

Response: This part of the manuscript has been thoroughly revised keeping in mind the advise of the reviewer.

The morphological differences between solitary spikelets and pseudospikelets have been elaborated in the result section (section 2.2, pages- 5-6, lines- 125-140).Please note that the basic inflorescence units (spikelet) and florets observed in solitary spikelets and pseudospikelets are quite similar. . Therefore, not much reproductive differences could be observed between florets obtained from solitary spikelets and pseudospikelets. However, their timing of appearance in a flowering cycle differed. Please see Figure 2 and section 3.1 (page- 13, lines- 312-324). Also, the rate of seed setting significantly varied between these two types of inflorescences (Figure 8D, section 2.6, page- 12, lines- 270-277 and section 3.2, page- 15, lines- 381-386). The SEM analysis on two types of inflorescence buds revealed formation of single apical inflorescence meristem (IM) in solitary spikelet (Figure 3D), whereas multiple inflorescence meristem (IM) arranged in a capitate manner in pseudospikelet (Figure 3H). This has been described in Results (section 2.2, pages- 5-6, lines- 131-132 and 138-140, yellow highlighted).

5. Discussion (line 261): I think "lowering" should be "flowering".

Response: We are sorry for the unintentional mistake. Now it has been corrected in the revised version of the manuscript. (page- 13, line- 309).

6. Discussion (lines 253-263): Sporadic vs. gregarious flowering patterns are compared, but what is the reproductive advantage of one against the other? Is it known or previously published?

Response: We thank Reviewer 1 for this valuable suggestion. The section 3.1 has now been revised based on this suggestion (page- 13, lines- 291-304, yellow highlighted). Gregarious flowering generally extends over a large area, while sporadic remain restricted in a few culm. Sporadic events may or may not be followed by the mass flowering. In mass flowering the rate of seed production is usually higher and flowering culms wither after each cycle, which was not the case for sporadic flowering. However, sporadic events, which are followed by gregarious flowering (sporadic-massive synchronize type) may have much severe impact on forest population.

7. Discussion (line 268): Phyllostachys aurea in italics.

Response: We thank Reviewer 1 for pointing this unintentional mistake. Now it has been  corrected in the revised version of the manuscript (page- 13, line- 316, yellow highlighted).

8. Discussion (lines 284-291): Here there is an important issue which should be discussed  profoundly. I am not experienced in bamboo species, but they appear insect-pollinated plants according to the next paragraphs. Taking into account the size of the pollen grain, it could be also dispersed by the air (Romero-Morte et al. 2018). Romero-Morte et al. compared different pollen sizes of grasses and the capability to be abundant in the air as aeroallergen. Clinical aspects do not appear to be one of the aims of this study, but it is important to discuss more about the pollination strategies of this species.

Response: We thank reviewer 1 for raising this important point and indeed, it was not properly explained in the earlier version of the manuscript. The respective paragraph has now been revised in the discussion section (section 3.2, page-14, lines- 355-366) and the suggested reference has been added in the updated version of the manuscript.

Bamboo is a member of the family Poaceae and is primarily wind pollinated. Although, presence of insect pollinators have been reported in a few species; no insect visitors were observed in this study. As per the suggested literature (Romero-Morte et al., 2018) B. tulda pollens fall into category 3. They host small pollens, which possess higher buoyancy in air resulting wind as the primary agent of pollen dispersal. Further, with respect to the clinical aspects, not much information is available on bamboo, although, majority grass pollens are known to be allergenic.

9. Discussion (lines 317-340): I really liked the way to demonstrate protandry and the reproductive advantage of cross-pollination in this species. Also, the results and the way of showing it in the manuscript.

Response: We thank the reviewer again for such encouraging comment.

10. Materials and methods (lines 386-426): Please, provide more information about how pollen grains are treated and collected, and if the authors used several replicates for each analysis to maintain the representativeness (Rojo et al. 2015). It should be clearly remarked in this section as it is relevant from a statistically point of view.

Response: The relevant section of the ‘Method’ related to pollen grains has been revised (section 4.4, page- 17-18, lines- 507-529). A brief description is provided below:

Multiple replicates were used for both TTC and in vitro germination analyses in order to test statistical significance. Three different populations of B. tulda (SHYM7, SHYM16, BNDL24) were selected for this study. For each population, florets were collected from three randomly selected culms between 8:00-9:00 AM during May, 2015 (~34-37°C). For each culm, three florets were obtained from randomly selected flowering branches. Thus, pollens obtained from total 162 anthers were used (3 populations X 3 culms X 3 florets X 6 anthers). In order to collect pollen grains, anthers from fresh florets were collected during post anthesis when the anthers were bright yellow or purple. Collectively 18 anthers obtained from 3 florets from each culm were pooled together and were subjected to TTC as well as in vitro germination assay. For pollen germination assay, pollens were immediately kept in micro centrifuge tubes containing ~1.0 ml of TTC solution, incubated in dark for 30 mins and observed under bright field microscope. The TTC solution was prepared by dissolving 0.5 gm of TTC to 100 ml of 15% sucrose solution. Total 2460 pollens were used for this analysis. On the other hand for in vitro germination analyses, pollens were immediately dusted in 0.5ml pollen germination medium (Brewbaker and Kwack’s medium). 

Reviewer 2 Report

The author describe by life and SEM images the reproductive organs of Bambusa tulda which is followed by pollen quality data, pollination experiments, and seed production. I consider the quality of the microscopic work as very good but mis details and explanations in several parts of the manuscript interpreting the results of pollen analysis and seed set. I don’t think it is correct using a different plants species as reference for pollen quality, and the data on seed set should be corrected for the number of flowers. The authors have shown in their microscopic analysis that the different inflorescences have not the same flower number.

In certain parts the style is not coherent (Latin names are not always in italic) plus some odd word choices and typing mistakes. The manuscript would benefit from being cross-checked by a native speaker.

I therefore recommend this manuscript to be accepted only after serious revisions.

Author Response

Reviewer 2

1. The author describe by live and SEM images the reproductive organs of Bambusa tulda, which is followed by pollen quality data, pollination experiments, and seed production. I consider the quality of the microscopic work as very good but miss details and explanations in several parts of the manuscript interpreting the results of pollen analysis and seed set. I don’t think it is correct using a different plants species as reference for pollen quality.

Response: We thank reviewer 2 for this suggestion and we agree with this. Overall, we now have included more detailed description on how experiments on pollen viability (section 4.4, pages- 17-18, lines- 507-529) and seed setting (section 4.6, pages- 18-19, lines- 572-579) were conducted. We also revised the section interpreting results and discussion on pollen viability (section 2.4, pages 8-9, lines- 211-227; section 3.2, pages- 13-14, lines- 329-340, yellow highlighted) and seed setting (section 2.6, page- 12, lines- 273-276).   

We agree with reviewer 2 and data on Oryza sativa used as a reference for pollen quality assessment has now been removed. The respective Method (section 4.4, page- 17, line- 506, grey struck though) Results (section 2.4, page- 9, lines- 218-221 and 227-230, grey struck through) and Figure 6 has been now updated.

2. The data on seed set should be corrected for the number of flowers. The authors have shown in their microscopic analysis that the different inflorescences have not the same flower number.

Response: We thank reviewer 2 for pointing out. Indeed, in the earlier version manuscript it was not properly explained and hence the misunderstanding might have arisen. The SEM analysis on two types of inflorescence buds revealed formation of single apical inflorescence meristem (IM) in solitary spikelet (Figure 3D), whereas multiple inflorescence meristem (IM) arranged in a capitate manner in pseudospikelet (Figure 3H). This has been described in Results (section 2.2, pages- 5-6, lines- 131-132 and 138-140, yellow highlighted).        

In order to compare the percentage of seed setting between solitary vs. pseudo-spikelet, seeds were counted from 96 solitary spikelets and 99 spikelet units obtained from 120 pseudo-spikelet. The solitary spikelet consists of a single spikelet unit, whereas , a pseudospikelet is comprised of cluster of multiple spikelet units (~3-34).  In order to avoid any confounding effect due to different number of spikelet units present in solitary vs. pseudospikelet on the number of total seeds observed, single spikelet unit having 7-9 florets were selected from each inflorescence type. The respective results (section 2.6, page- 12, lines- 273-276, yellow highlighted) and materials and methods (section 4.6, pages- 18-19, lines- 572-579, yellow highlighted) sections has been revised in the updated version of the manuscript.

3. In certain parts the style is not coherent (Latin names are not always in italic) plus some odd word choices and typing mistakes. The manuscript would benefit from being cross-checked by a native speaker.

Response: The manuscript has been revised as per suggestion. Now the language has also been carefully edited by Professor James H. Westwood, School of Plant and Environmental Sciences, Virginia Tech., USA.

Reviewer 3 Report

It was interesting to read this article, it is well written, touches on many aspects of bamboo flowering, and has quality illustrations. In addition, the researchers did a great job observing bamboo populations and their blooms for 7 years. I think this article will be of interest to readers. Despite the overall favorable impression, several comments and recommendations can be made, mainly regarding the design and details of the text.

  1. There are four flowering modes listed in the introduction, but only two in the results and discussion. It's a bit confusing, why are the other two needed (sporadic + massive and partial)? How can we talk about sporadic flowering when there is already massive synchronous flowering (type 3) and what then is type 4?
  2. Table 1 would be more convenient for viewing if it had drawn cells. I would also shrink columns 1 and 2 so that there is more room in column 3. Then the information would be perceived more easily.
  3. After considering Figure 2, the question arose about the survival of an additional shoot in a flowering clump. Does it wither too? It looks like this in the picture. I found the answer to my question in the Discussion, apparently, it survives. Please clarify this point in the picture or in the text of the Results.
  4. In Figure 3, it would be great to bring the diagrams of the inflorescences next to the photo, this would increase the visual accessibility of the information.
  5. Line 191 - the decoding of TTC should be in the text, and not only in the figure and in the methods, since it occurs earlier in the text.
  6. I was a little confused by the expression 6.00 / 10.00 hour, it seems to me that it would be more familiar to just 6.00 or 6 AM
  7. As a pollen specialist, I was very interested in this low quality of pollen. It seems to me that it would be worthwhile to discuss in more detail the reasons for such a problem, which, together with the spatial division of the population, apparently leads to a low efficiency of seed formation. Could it be that the low observed quality of pollen is caused by some methodological reasons associated with the collection of material and the assessment of its viability? By the way, the percentage of sucrose in the medium seemed strange to me, 25% is a lot. I recently reviewed an article that discussed the optimal sucrose content in media for different species, usually much lower (10-18%) https://doi.org/10.3389/fpls.2021.709945
    On the other hand, could it be that the quality of pollen has decreased due to anthropogenic impact on these territories?

Author Response

Reviewer 3

1. There are four flowering modes listed in the introduction, but only two in the results and discussion. It's a bit confusing, why are the other two needed (sporadic + massive and partial)? How can we talk about sporadic flowering when there is already massive synchronous flowering (type 3) and what then is type 4?

Response: We thank Reviewer 3 for this valuable suggestion. . We now have described all the four types of flowering in Introduction (page- 2, lines- 50-65). The sporadic flowering occur in few culms of a population, whereas, gregarious flowering happens in a synchronized manner over a large geographical area. Quite often, sporadic events may be followed by mass flowering in the subsequent years and is defined as sporadic followed by massively synchronized flowering (Type 3). The type 4 is Partial flowering, which take place in small, discrete populations.  It is neither extended like gregarious, nor restricted like sporadic type with respect to the number of culms flowered. Like mentioned in the Introduction, we now have described the four above-mentioned flowering type in Discussion as well (section 3.1, page- 13, lines- 291-304, yellow highlighted).

2. Table 1 would be more convenient for viewing if it had drawn cells. I would also shrink columns 1 and 2 so that there is more room in column 3. Then the information would be perceived more easily.

Response: We thank reviewer 3 for this suggestion. The Table 1 has been now revised as per suggestion in the updated version of the manuscript.

3. After considering Figure 2, the question arose about the survival of an additional shoot in a flowering clump. Does it wither too? It looks like this in the picture. I found the answer to my question in the Discussion, apparently, it survives. Please clarify this point in the picture or in the text of the Results.

Response: We thank Reviewer 3 for pointing this out. We have included it in the Result (section 2.1, page- 5, lines- 120-122) and also the revised Figure 2 has been incorporated. In the event of sporadic flowering in B. tulda, rhizomes of the flowering clump remained active and young culms sprouted from the rhizomes. These sprouted culms attained maximum height before winter.

4. In Figure 3, it would be great to bring the diagrams of the inflorescences next to the photo, this would increase the visual accessibility of the information.

Response: We again thank the reviewer for this suggestion. The Figure 3 has been now revised as per the suggestion provided. .

5. Line 191 - the decoding of TTC should be in the text, and not only in the figure and in the methods, since it occurs earlier in the text.

Response: We thank reviewer 3 for pointing out this unintentional mistake. The full form of TTC has now been included in the Results (section 2.4, page- 9, line- 215, yellow highlighted).

6. I was a little confused by the expression 6.00 / 10.00 hour, it seems to me that it would be more familiar to just 6.00 or 6 AM

Response: We thank reviewer 3 for pointing out this confusion. The expression of hour has been now revised by including AM or PM.

7. As a pollen specialist, I was very interested in this low quality of pollen. It seems to me that it would be worthwhile to discuss in more detail the reasons for such a problem, which, together with the spatial division of the population, apparently leads to a low efficiency of seed formation. Could it be that the low observed quality of pollen is caused by some methodological reasons associated with the collection of material and the assessment of its viability? By the way, the percentage of sucrose in the medium seemed strange to me, 25% is a lot. I recently reviewed an article that discussed the optimal sucrose content in media for different species, usually much lower (10-18%) https://doi.org/10.3389/fpls.2021.709945
On the other hand, could it be that the quality of pollen has decreased due to anthropogenic impact on these territories?

Response: We thank reviewer 3 for this helpful comment.

We also agree with the suggestions that the spatial distribution of the populations may result into lower seed set. This notion has now been incorporated in the Discussion (section 3.2, page- 14, lines- 368-378) of the revised manuscript.  A larger flowering area implies higher load of pollen dispersal and availability. In contrast, the studied B. tulda populations (BNDL23, SHYM7 and SHYM16) were small and isolated, resulting into low availability of pollen and consequently low seed setting.

With respect to the issue on pollen quality associated with methodologies used in this study, we would like to indicate that although sufficient precautions were undertaken to maintain optimum condition during pollen collection and assay, yet there could be possibilities of technical errors like time to transport them to the laboratory and small sized population frequently subjected to anthropogenic impact, which were beyond our control (section 3.2, pages- 13-14, lines- 329-339, yellow highlighted). Since there are previous observations in bamboo that chromosomal aberration might have caused low pollen germinability (4.5%), we have done additional experiments to test this in B. tulda. Therefore, meiotic behavior of PMC has been studied (Figure S1).   .

We also thank the reviewer for pointing out the unintentional typographical error regarding the concentration of sucrose used in germination medium for in vitro pollen viability assay. The actual concentration of sucrose used was 15%, not 25%. In order to identify the ideal concentration of sucrose for optimum germination, experiments were conducted using media supplemented with 10, 15, 20, 25 and 30% sucrose (w/v). The optimum percentage of germination was observed in media supplemented with 15% sucrose and hence was used for in vitro germination assay. This data has now been incorporated as Table S1. All these relevant information have now been provided in Methods (section 4.4, pages- 17-18, lines- 524-529) and Results (section 2.4, page- 9, lines- 224-225, yellow highlighted).    

Reviewer 4 Report

Thank you for the opportunity to evaluate this manuscript on aspects of the reproductive biology of Bambusa tulda.  I certainly learned more about bamboo reproduction.  The study is very descriptive, and while there are some aspects of comparison to other species of grasses in the Discussion, I unfortunately find the study to be lacking greater context.  For example, according to Langridge and Baumann (2008), there are no reports of SI in Bambusoideae, and Poaceae has a unique type of SI controlled by two, unlinked loci, so it would be nice to see an expansion on the reason(s) there might be low seed set (and low pollen viability).  Additionally, are the populations studied able to exchange pollen?  How do the results between the two pairs of geographically close populations compare to those that are geographically distant?  Are there reasons that these types of comparisons weren't made more explicitly?  I'm all for descriptive studies, rather than those that might be more hypothesis driven, but I think the authors have an opportunity to expand their study to be of greater interest to the botanical community rather than just a descriptive study on some areas of reproductive biology of one bamboo species.  Finally, the English language and grammar should be improved.

Author Response

Reviewer 4

1. According to Langridge and Baumann (2008), there are no reports of SI in Bambusoideae, and Poaceae has a unique type of SI controlled by two, unlinked loci, so it would be nice to see an expansion on the reason(s) there might be low seed set (and low pollen viability).

Response: We thank reviewer 4 for pointing out this important issue related to SI and its possible impact on seed setting. Now this aspect has been included in the Discussion (section 3.3, page- 15, lines- 415-427, yellow highlighted).

Since both S and Z loci are multi allelic, their allelic diversity can be an important factor that determine the interaction between pollen and pistil during a compatibility reaction. In case of cross pollinated species, self pollination may occur due to unavailability of compatible pollen grains. This may result into rejection of maximum pollens due to lower allelic diversity during their interaction with pistil. This may lead to lower rate of fertilization and consequently low seed set. Although the two loci based gametophytic SI (GSI) is known in grasses, information on the genes and their regulation are scanty and in Bambusoideae they are even rarer. Since diverse mating behavior is observed in bamboos and an individual species can determine its mating nature based on availability of pollen grains, pollinators and environmental factors, it would be interesting to study if similar S-Z loci based GSI system exists in bamboo and extent of their allelic variation caused by polyploidization.

2. Additionally, are the populations studied able to exchange pollen?  How do the results between the two pairs of geographically close populations compare to those that are geographically distant?  Are there reasons that these types of comparisons weren't made more explicitly?

Response: We thank reviewer 4 for pointing out this important issue, which has now been discussed in the Method (section 4.5, page- 18, lines- 540-545).

Even though flowering was studied in four different populations- SHYM7, SHYM16, BNDL23 and BNDL24, the artificial pollination experiment was performed only in BNDL23 and BNDL24. The SHYM7 and SHYM16 population pairs are ~630m distant from each other, whereas BNDL23 and BNDL24 are ~340m distant from each other (Figure 1). But the distance between BNDL pair and SHYM pair are ~33km. Even though the BNDL23 and BNDL24 are relatively closer, they are ~350m distant from each other, which is longer than the optimum dispersal distance (~55m) as observed in maize (Bannert, 2007). Thus, the probability of natural pollen exchange between these two populations are quite low. On the other hand, pollination experiment could not be performed between the two pairs of geographically distant populations, as there is high possibilities of reduction in pollen viability during transportation due to the long distance between them. In collaboration with Dr. Uday Chatterjee, we now have calculated the geographical distance between all the populations studied and have been incorporated them in the Revised Figure 1.

3. I'm all for descriptive studies, rather than those that might be more hypothesis driven, but I think the authors have an opportunity to expand their study to be of greater interest to the botanical community rather than just a descriptive study on some areas of reproductive biology of one bamboo species.

Response: We thank reviewer 4 for providing such an useful advise to improve the overall quality of the manuscript. Keeping this in mind, we now have revised the manuscript all throughout. For instance, the GSI mechanism of grasses have now been discussed and compared to that of bamboo on the basis of available information (section 3.3, page- 15, lines- 415-427).

4. Finally, the English language and grammar should be improved.

Response: Now the language has been crosschecked all throughout the manuscript by Professor James H. Westwood, School of Plant and Environmental Sciences, Virginia Tech, USA.

Round 2

Reviewer 2 Report

Dear authors, the manuscript has improved a lot. A few things that I saw: in Figure 4G, the abbreviation for "S" is not given in the legend. The arrow is pointing at the stigma but than might not be obvious for everyone.

In Supp Table 1, pollen has no plural, thus no "s" at the end.